# Heme Oxygenase-1 Deficiency and Oxidative Stress: A Review of 9 Independent Human Cases and Animal Models

**DOI:** 10.3390/ijms22041514

**Published:** 2021-02-03

**Authors:** Akihiro Yachie

**Affiliations:** Division of Medical Safety, Kanazawa University Hospital, Kanazawa 920-8641, Japan; yachie@staff.kanazawa-u.ac.jp; Tel.: +81-76-265-2073

**Keywords:** heme oxygenase, HO-1 deficiency, oxidative stress, inflammation

## Abstract

Since Yachie et al. reported the first description of human heme oxygenase (HO)-1 deficiency more than 20 years ago, few additional human cases have been reported in the literature. A detailed analysis of the first human case of HO-1 deficiency revealed that HO-1 is involved in the protection of multiple tissues and organs from oxidative stress and excessive inflammatory reactions, through the release of multiple molecules with anti-oxidative stress and anti-inflammatory functions. HO-1 production is induced in vivo within selected cell types, including renal tubular epithelium, hepatic Kupffer cells, vascular endothelium, and monocytes/macrophages, suggesting that HO-1 plays critical roles in these cells. In vivo and in vitro studies have indicated that impaired HO-1 production results in progressive monocyte dysfunction, unregulated macrophage activation and endothelial cell dysfunction, leading to catastrophic systemic inflammatory response syndrome. Data from reported human cases of HO-1 deficiency and numerous studies using animal models suggest that HO-1 plays critical roles in various clinical settings involving excessive oxidative stress and inflammation. In this regard, therapy to induce HO-1 production by pharmacological intervention represents a promising novel strategy to control inflammatory diseases.

## 1. Background

Heme is a major component of hemoglobin (Hb), and is a product of erythrocyte destruction. Heme is constantly produced in vivo through the destruction of both senescent erythrocytes under physiological conditions, and abnormal erythrocytes under pathological conditions. Heme is extremely toxic to cells, so constitutive mechanisms exist to cancel the toxic effects of heme [1,2]. Serum haptoglobin (Hp) binds to free Hb efficiently [3,4] and the resulting Hb-Hp complex is promptly taken up by phagocytes and hepatocytes, which express a receptor for the complex that is currently known as CD163 [5,6,7]. Heme oxygenase (HO)-1 constitutes one of the three isozymes of HO, and catalyzes the degradation of heme into biliverdin, carbon monoxide (CO), and free iron, each of which somehow exert potent anti-oxidative stress and anti-inflammatory functions [8,9]. For this reason, HO-1 exerts protective properties in a broad range of human diseases, including neurodegenerative diseases, cardiovascular diseases, cancer, metabolic diseases, iron metabolism disorders and various inflammatory diseases [10] (Figure 1).

Among the heme degradation products, CO plays a central role and acts on cellular metabolisms to protect cells from oxidative stress and regulate the production of inflammatory molecules [11,12,13]. CO directly controls the inflammatory state of a given tissue, and at the same time regulates the microcirculation within target organs by acting as a gaseous mediator [14,15]. Among the three isozymes, HO-1 is the only protein rapidly induced upon stimulation with various oxidative stresses [14,16]. Therefore, any defect in the function of HO-1 would be expected to lead to uncontrollable inflammation in response to certain exogenous insults, such as infection and hemolysis.

After Yachie et al. reported the first human case of HO-1 deficiency [17,18], 5 cases with identical mutation were recognized in India [19,20,21]. Additional cases with nonsense mutations and a case with a missense mutation have recently been reported [22,23,24]. In this short review, I describe the clinical characteristics and laboratory features of these cases and propose denominators for this extremely rare genetic disease.

## 2. Case Reports of Human HO-1 Deficiency

Although numerous studies have been published utilizing animal models and in vitro experiments with regard to the role of HO-1 in vivo, only a limited number of human cases have been reported to date. Although the number of such cases remains limited, clinical descriptions of these human cases of HO-1 deficiency highlight the critical roles played by this enzyme in protecting cells and organs from oxidative stress and inflammation.

### 2.1. The First Case

In 1999, Yachie et al. reported on a 26-month-old boy who had been admitted with recurrent fever, generalized erythematous rash, and joint pain [17]. The combination of fever, erythematous rash, and joint pain suggested that the patient suffered from a childhood chronic inflammatory illness such as systemic juvenile idiopathic arthritis (sJIA) or chronic infantile neurological cutaneous and articular syndrome (alternatively called neonatal-onset multisystem inflammatory disease). The latter condition is now known as cryopyrin-associated periodic syndrome. However, neither of these conditions could explain the unique clinical features and extraordinary laboratory findings seen in this patient. The patient had characteristic facial features such as a flat nasal bridge, frontal bossing, and prominent edema of the eyelids. Although the liver was markedly enlarged, the spleen was absent. Unlike most patients with asplenia, this patient did not have any form of congenital heart disease. His brother and sister appeared healthy, but the mother had experienced two intrauterine fetal deaths.

The most significant characteristics of his laboratory data were prominent increases in white blood cell count (51,600/μL) and platelet count (226 × 10^4^/μL). This marked leukocytosis and thrombocytosis persisted throughout the course of the illness. The patient showed significant anemia with an erythrocyte count of 1.48 × 10^6^/μL and a hemoglobin concentration of 4.9 g/dL. Peripheral blood smear showed numerous fragmented erythrocytes, Howell-Jolly bodies and nucleated red blood cells. Serum iron concentration was normal at 64 μg/dL. The level of lactate dehydrogenase (LDH) was extremely elevated (17,470 IU/L, normal range; 196–355) and aspartate aminotransferase (AST) was also increased (442 IU/L, normal range; 9–42), but alanine aminotransferase (ALT) remained within normal limits. Serum ferritin was 780 ng/mL (normal range; 26–280). Marked abnormalities were noted in parameters of both the coagulation and fibrinolysis systems, although he did not show any apparent bleeding tendency or signs of accelerated coagulation. Fibrinogen was 109 mg/dL (normal range; 196–356), fibrin degradation product was 300.1 μg/mL (normal range; <5), d-dimer was 186.1 μg/mL (normal range; <2.5), thrombin-antithrombin complex was 202.2 μg/L (normal range; <3), and plasmin-2 plasmin inhibitor complex was 22.3 μg/mL (normal range; <0.8). Thrombomodulin was 12 ng/mL (normal range; <3.5) and von Willbrand factor was 580% (normal range; 32–115). Hyperlipidemia was another prominent finding, with triglycerides at 638 mg/dL (normal range; 32–115) and total cholesterol at 552 mg/dL (normal range; 128–219), showing a predominance of low-density lipoprotein cholesterol.

Serum specimens looked quite peculiar, appearing turbid with a brownish tint (Figure 2A). This was not due to mechanical damage to the erythrocytes through any traumatic procedures, or the result of ex vivo destruction of fragile erythrocytes. Repeated blood sampling yielded similar results. Venous blood samples were spun down right after venipuncture and serum always appeared the same. Given the dark coloration of the patient’s serum samples, certain hemoglobinopathies were suspected. Hemolysate of the erythrocytes was transparent and showed only two distinct peaks corresponding to oxyhemoglobin (OxyHb), at 541 nm and 576 nm (Figure 2B). In addition to these two peaks for OxyHb, serum from the patient showed a third unique peak at 631 nm, corresponding to methemoglobin (MetHb). The gross appearance of the serum combined with the results of absorption spectrum analysis suggested that heme in the form of OxyHb and MetHb was markedly increased in the patient’s serum (Figure 2C).

These results indicated that either massive hemolysis was constantly taking place in vivo or that Hb was accumulating in serum due to defects in the Hb catabolic pathway. Further study revealed that serum Hp concentration was extremely elevated (800–1200 mg/dL, normal range; 40–180 mg/dL) and a large amount of Hb-Hp complex was detected in a urine sample from the patient. These data suggested that the Hb-Hp complex was somehow bypassing the normal scavenger system and overflowing into urine. Repeated measurements of serum bilirubin concentration were always low, at 0.1–0.3 mg/dL. Hemopexin was undetectable by immunoelectrophoresis, indicating the presence of active hemolysis in vivo. Serum heme concentration was extremely high, at 490 μM (normal range; <1 μM). Results for both direct and indirect Coombs’ tests were negative on repeated occasions. All these data together were highly suggestive of abnormalities in either the process of hepatic uptake of Hb-Hp complex or the heme degradation pathway (Figure 3).

Immediately before we reported this case of human HO-1 deficiency, Poss and Tonegawa had made monumental reports describing the characteristics of HO-1-deficient mice [25,26]. Those results shared many findings in common with our patient, including severe anemia due to defective iron reuse, tissue iron deposition, enhanced cell injury due to oxidative stress and accelerated tissue inflammation. Immunohistochemical analysis of the liver biopsy specimen showed that Kupffer cells did not produce HO-1 in the patient’s liver. Exposure of the patient’s monocytes to heavy metals, such as cadmium or sodium arsenite, did not induce HO-1 protein. Immunoblotting analysis revealed that lymphoblastoid cell lines derived from the patient did not produce HO-1 protein in response to these oxidative stressors. *HMOX1* analysis revealed that the patient was a compound heterozygote for two *HMOX1* mutations: the maternal allele lacked the 2nd exon, while the paternal allele showed a 2-base pair deletion within the 3rd exon.

Pathological examination of the first case of HO-1 deficiency revealed characteristic tissue injury [18]. Notably, cellular injury was confined to selected organs and cell types, including the kidneys, liver, circulating monocytes, and vascular endothelial cells. In the kidneys, mild mesangial proliferation and thickening of the capillary loop were observed within glomeruli. Electron microscopy revealed marked swelling of the endothelial cells and detachment throughout the glomerular capillaries. In addition to glomerular damage, tubulointerstitial injury with tubular atrophy was significant. The liver was massively enlarged and significant amyloid accumulations had resulted in marked atrophy of hepatocytes. Scattered foci of iron deposits were observed in both the kidneys and liver. The majority of iron deposits were detected within renal tubular epithelial cells and hepatic parenchymal cells. The cytoplasm of circulating monocytes was vacuolated, and expressions of many monocyte surface antigens were markedly reduced. Although we could not see whether circulating monocytes from the patient expressed Hb-Hp receptor CD163, no expression was detectable on Kupffer cells from the liver of the patient [27].

### 2.2. The Indian Cases: 5 Independent Cases with an Identical HMOX1 Mutation

After we reported the first case of human HO-1 deficiency, 5 cases from India were confirmed by HO-1 gene analysis (Radhakrishnan 2011 [19], Radhakrishnan 2011 [20], Gupta 2016 [21], and personal communications). Fever, absence of the spleen, hemolytic anemia, hematuria/proteinuria, and the absence of jaundice seem to be the common findings in those cases. All cases from India (Cases 2–6) displayed an identical homozygous mutation (p.R44X), indicating a founder effect of this particular mutation in that country. Although no functional study has been performed to confirm the impact of this particular mutation, it is a nonsense mutation and no HO-1 protein was detectable within the tissues examined. This mutation was thus suggested to be deleterious, resulting in impaired control of oxidative stress and inflammation. However, ages at disease onset and durations varied significantly between cases. Age at onset varied from 6 months to 15 years. Surprisingly, most patients remained asymptomatic for a long time until the onset of the first recognizable symptom. Case 2 (the first of the 5 Indian cases) remained healthy until she experienced prolonged fever at 15 years old. Growth and development were both normal and she had no significant past history. However, her condition deteriorated rapidly once the disease became apparent. Her terminal stage was complicated by hypertension and intracranial hemorrhage, as observed in the initial Japanese case. She died 5 months after the diagnosis, when fungal sepsis developed. Furthermore, two other reported cases, Case 3 and Case 6 also showed rapid deterioration and died soon after the onset of symptoms. The variability of age of onset and the rapidly progressive clinical course suggests the presence of certain compensatory mechanisms to overcome the absence of HO-1 function. The devastating clinical courses experienced by these patients indicate the critical importance of HO-1 in holding back rapidly progressive inflammation and organ dysfunction once an overwhelming inflammatory response is ignited by certain triggers.

In all 5 Indian cases, laboratory findings were surprisingly similar, characterized by markedly increased platelet counts, leukocytosis and significant anemia. Levels of hepatic enzymes such as LDH, AST, and ALT uniformly showed significant elevations. Low to normal serum bilirubin and high haptoglobin concentration in the presence of hemolysis seem to be hallmarks of the illness. In addition to constantly elevated C-reactive protein (CRP) values, significant levels of serum ferritin were another characteristic of these cases. The cases also shared many clinical characteristics. However, clinical profiles were more variable than laboratory findings. Although asplenia and a prominent forehead were observed in all 5 cases, growth delay was not seen in Case 2 and Case 6. Hypertension was not seen in Case 6. Cerebral bleeding was observed only in Case 2 and Case 3. Arthralgia was seen in the first case, but was not detected in any of the 5 Indian cases.

### 2.3. The Iranian Case: A Novel HMOX1 Mutation

Tahghighi et al. reported a 7th case of HO-1 deficiency, in a girl who was 17 months old when she presented with high fever, tachypnea, and respiratory distress [22]. She was the third child born to consanguineous Iranian parents. The first child in the proband’s family was a girl who was found to be completely healthy, but their second child was aborted spontaneously at 20 weeks for unknown reasons. On first admission, the patient revealed massive pericardial effusion without any evidence of infectious diseases or malignancies. Hepatomegaly was found and a liver biopsy showed iron deposits within the hepatic parenchyma. Laboratory findings included leukocytosis, increased platelet count, hemolytic anemia, and elevated levels of inflammatory markers. Increased levels of hepatic enzymes, triglycerides and ferritin were also noted. In addition to infectious diseases and malignancies, autoinflammatory diseases or immunodeficiencies were ruled out based on the clinical and laboratory findings. Corticosteroid proved ineffective for controlling inflammation and her condition deteriorated progressively. She died of recurrent fever, bleeding, heart failure, and ascites after 4 admissions. Due to the clinical profile and characteristics of laboratory data, Tahghighi et al. suspected hemophagocytic lymphohistiocytosis (HLH) or HO-1 deficiency. Post-mortem whole-exome sequencing revealed homozygous mutations of the *HMOX1* gene (within exon 3, p.K204X). Both parents carried identical mutations on a single allele of the *HMOX1* gene.

Although the patient had a normal sized-spleen, other clinical and laboratory features were similar to those found in the Japanese (Case 1) and the 5 Indian cases (Cases 2–6). The characteristic laboratory features included leukocytosis, increased platelet count, anemia, elevated CRP, extremely high LDH and ferritin concentrations, hyperlipidemia and low to normal bilirubin levels. Clinical findings included prolonged or recurrent fever, hepatomegaly and hemolytic anemia. Although Tahghighi et al. did not perform any functional studies to confirm that the specific mutation resulted in defective enzyme function, ENTPRISE-X (http://cssb2.biology.gatech.edu/entprise-x/) predicted that mutations at codon 204 of HO-1 would be deleterious. A p.K204X mutation produces a truncated protein of 203 amino acids (instead of 288 amino acids) that is predicted to be highly pathological.

The reason the patient with K204X mutation had a normal-sized spleen remains unclear. Tahghighi et al. did not mention if postmortem pathological examinations were performed to identify any structural abnormalities within the patient’s spleen. If HO-1 deficiency has any effect on splenic development, as suggested by the previous human cases and the findings from HO-1-knockout mice, the spleen would be highly expected to show significant pathological findings reflecting impaired splenic vascularity.

### 2.4. The Turkish Case: A Missense Mutation

The case reported by Greil et al. (Case 8) represented the first case of missense mutation in the *HMOX1* gene (p.G139V) [23]. The patient was the son of consanguineous Turkish parents. Clinical profiles were characteristic, in that he presented with microcytic anemia and progressive hepatosplenomegaly at 3 months old. The anemia proved resistant to iron supplementation and the patient was dependent on red cell transfusion. Liver biopsy revealed severe hemophagocytosis and Kupffer cell siderosis with extramedullary hematopoiesis. Slight hemophagocytosis was also seen in the bone marrow. Because of these clinical findings and the laboratory data, HLH was suspected and the patient received treatment with immunochemotherapeutic agents (HLH2004 protocol) with sustained remission of the signs of HLH [28]. However, elevated levels of inflammatory markers, including interleukin (IL)-1β, IL-6, tumor necrosis factor (TNF)-α, ferritin, and CRP remained, indicating the sustained presence of an intense inflammatory reaction.

DNA sequence analysis was performed to identify any known gene mutations responsible for familial HLH, but failed to find any. Persistently elevated LDH despite low bilirubin levels led Greil et al. to analyze the *HMOX1* gene, revealing the homozygous G139V mutation. This mutation was a missense mutation and expression of constitutive HO-1 was increased by cells, but HO-1 levels were not enhanced with additional oxidative stress. Apoptosis of peripheral blood mononuclear cells was examined upon exposure to oxidative stress (heme) and a significant difference in the level of cell death was seen between control and the patient’s cells, suggesting a functional defect of HO-1 activity in the patient’s cells.

Although many features were similar to previously described cases of HO-1-deficient patients, some findings were distinct and not observed in other patients. Whereas HO-1 enzyme activity was significantly decreased, the mutation was associated with a gain of abnormal peroxidase function of the enzyme, resulting in increased urinary excretion of peroxidation products. Furthermore, mononuclear cells from the patient produced significantly high concentrations of inflammatory cytokines, such as TNF-α and IL-6, upon stimulation with lipopolysaccharide (LPS). These unique functional characteristics of mutant HO-1 resulted in distinct clinical features of this particular patient compared to other reported cases of human HO-1 deficiency with null-type mutations. Of particular interest is the fact that the patient with G139V mutation developed paradoxical inflammatory response to red cell transfusion, presumably due to abnormal peroxidase function provoked by constitutively enhanced expression of the mutated HO-1 product.

### 2.5. The Case from USA: A Compound Heterozygote of Paternal Frameshift and Maternal Splice Donor

The most recent report of HO-1 deficiency by Chau et al. (Case 9) was characterized by interstitial lung disease associated with frequent flares of systemic inflammation [24]. The patient presented with the first inflammatory episode at 4 years of age with a one-month history of fatigue, intermittent fevers, and dark urine. Mild prognathism and slight frontal prominence were noted. High fever and hypoxemia persisted during hospitalization. The laboratory examination revealed leukocytosis, thrombocytosis, elevated LDH at 19,706 U/L, and hyperferritinemia to 1980 ng/mL. Hemolytic anemia with schistocytes and Howell-Jolly bodies were noted on peripheral blood smears. There was no evidence of autoimmune hemolytic anemia. Serum bilirubin level was normal. Abdominal CT scan revealed poorly perfused, hypoplastic spleen. Hepatomegaly was another characteristic finding and the liver biopsy revealed mild sinusoidal fibrosis, mild microvesicular steatosis, and increased iron deposit within Kupffer cells.

His growth had been normal by 3 years of age. However, his growth slowed down significantly after the onset of the inflammatory episodes, being less than the 10th percentile for weight and 20th percentile for height at 8 years. Because of persistent and progressive respiratory symptoms, a biopsy of the lung was performed at 10 years and it demonstrated extensive fibrotic nonspecific interstitial pneumonia, patchy pleural fibrosis, and scattered cholesterol granulomas. Flares of inflammatory episodes were associated with fevers, respiratory symptoms and laboratory evidence of hemophagocytosis, suggesting macrophage activation syndrome or HLH. Because of these clinical characteristics, genetic testing for periodic fever syndromes and familial HLH was performed. However, no pathogenic mutation was identified in known genes. Therapy with corticosteroid, anti-IL-1R, anti-IL-6, and cyclosporine had been tried but with minimal benefit. He died at 10 years of age due to respiratory failure.

Post-mortem whole exome sequencing was performed on the patient and the family members. It revealed that the patient had a compound heterozygous mutations of *HMOX1*, with paternal frame shift c.264_269delCTGG (p.L89Sfs*24) and maternal splice donor c.636 + 2T > A, consistent with HO-1 deficiency. The patient peripheral blood mononuclear cells did not produce HO-1 upon stimulation with cobalt protoporphyrin.

Compared with the rest of the HO-1 deficiency cases, Case 9 is characterized by chronic and progressive lung disease. The pathological finding of the lung, pulmonary interstitial and intra-alveolar cholesterol granulomas, is a very rare finding. Chau et al. regard this as the sequela of pulmonary manifestation of small vessel vasculitis and macrophage activation. This case shows further phenotype expansion for HO-1 deficiency and at the same time, indicate the critical role of HO-1 in preventing inflammation and macrophage activation in various tissues and organs.

### 2.6. Denominators of Human HO-1 Deficiency

Since Yachie et al. reported the first HO-1 deficiency patient, eight additional cases of HO-1 deficiency with nonsense or missense *HMOX1* gene mutations have been described. A comparison of clinical characteristics is shown in Table 1 and the laboratory features of these cases are shown in Table 2. Why so few patients have been reported so far remains unclear. HO-1 may play cardinal roles during fetal growth and development, and most individuals with HO-1 deficiency may thus develop significant organ dysfunctions and die in utero or shortly after birth without being diagnosed. Unless the distinct clinical and laboratory features are recognized in time, thinking of HO-1 deficiency as a candidate diagnosis for the patient is extremely difficult, especially when the condition is still largely unfamiliar to clinicians. For this reason, proposing denominators of human HO-1 deficiency and recognizing this rare and difficult-to-treat disease early during the course of illness is of paramount importance.

Although ages at onset among the 9 HO-1 deficiency patients were quite variable, ranging from infancy to 15 years of age, laboratory data and clinical profiles were surprisingly uniform. Fever and hemolytic anemia were constant findings. Hematuria and proteinuria were observed in all of the Japanese and Indian cases. The reports on Cases 7 and 8 failed to mention whether the patients showed abnormal urine findings. The presence of hematuria or proteinuria would support the possibility of kidney injury in HO-1 deficiency. More importantly, bilirubin remained low or within normal ranges in all cases, despite the presence of active hemolytic anemia. Paradoxical normo- or hypobilirubinemia in the presence of active hemolytic anemia is certainly the first denominator of HO-1 deficiency.

Another distinct laboratory feature is the extremely elevated serum ferritin and LDH values. Although Cases 7, 8, and 9 showed at least transient episodes of HLH during the course of the illness, which may partly explain the high ferritin and LDH values, the clinical features were incompatible with typical HLH cases. More importantly, increases in leukocyte and platelet counts were invariably noted in all cases, which is contrary to the phenomenon observed in HLH patients. Extremely high ferritin and LDH values in the absence of apparent features of HLH may thus be the second denominator of HO-1 deficiency.

Of particular importance, absence or hypoplasia of the spleen seems to be the hallmark of the illness, at least for the first 6 cases. Absence of the spleen seemed to be a specific finding for the first 6 HO-1-deficient cases and Case 9, but was not the case in Cases 7 and 8. Why these two cases showed a normal-sized spleen or splenomegaly, respectively, remains unclear. In this regard, a report by Kovtunovych et al. showed HO-1 mutation in mice resulting in progressive atrophy of the spleen due to fibrotic changes in the splenic artery with age, providing food for thought [29]. Although not described in the reports, knowing whether Cases 7 and 8 showed any evidence of splenic functional deficit would be of interest. Asplenia is certainly the most important denominator of human HO-1 deficiency if recognized in the absence of congenital heart disease. However, the possibility of HO-1 deficiency should not be excluded because the patients showed a normal-sized or enlarged spleen.

If cases are accumulated in the future, a characteristic facial appearance may be able to be defined, but at present it is difficult to regard a prominent forehead or eyelid edema as a uniform characteristic of HO-1 deficiency patients. Hypertension and intracranial hemorrhage are certainly related to the basic pathology of HO-1 deficiency. However, these symptoms were not constantly observed in HO-1 deficiency patients. Information must be collected from a large number of cases to extend the list of denominators for the correct diagnosis of patients.

## 3. Animal Models of HO-1 Deficiency

Poss and Tonegawa reported on HO-1-knockout mice with a C57BL/6 background in 1997, showing that these animals lacked the ability to reuse iron and were characterized by progressive anemia, tissue iron deposition, chronic inflammation, and delayed growth [25,26]. Serum iron levels were abnormally low. In contrast, there was intense accumulation of iron within the renal tubular epithelium and hepatic parenchyma, contributing to macromolecular oxidative damage, tissue injury, and chronic inflammation. Progressive increases in serum ferritin were seen along with progression of anemia and leukocytosis. Although spleen size was normal at birth, marked enlargement of the spleen developed as mice aged. Furthermore, these mice were extremely sensitive to oxidative injury and were prone to death with LPS administration. Embryonic fibroblasts derived from HO-1-knockout mice showed increased sensitivity to oxidative stress, such as hemin, H_2_O_2_, paraquat, and heavy metals. All these results indicated that HO-1 is important for iron homeostasis and indispensable for the rapid protection of cells and tissues from potential oxidative damage during stress.

Kovtunovych et al. published a detailed study on the iron distribution and pathology of another strain of HO-1-knockout mice using a mixed C57BL/6-FVB background [29]. They showed progressive death of macrophages in the liver and spleen throughout the process of erythrophagocytosis and the resultant heme release in vivo, causing significant damage to the organs and intense inflammation of the surrounding tissues. In contrast to the splenomegaly seen in HO-1-knockout mice reported by Poss and Tonegawa, the spleen in these mice showed initial splenomegaly when young, but progressive atrophy as the mice aged. Pathological examination revealed initial enlargement of the red pulp of the spleen underwent progressive fibrotic changes and eventual atrophy and functional hyposplenism in older mice. Impaired splenic function was confirmed by the appearance of Howell-Jolly bodies within erythrocytes. As seen in human HO-1 deficiency patients and in the first strain of HO-1-knockout mice, iron deposition was prominent within the renal proximal tubules and hepatic parenchymal cells. Reduction of the Hb-Hp scavenger receptor, CD163, by macrophages was also noted, further indicating dysfunction of macrophages as the important producers of HO-1.

A rat model of HO-1 deficiency was recently described by Atsaves et al. [30]. Sprague-Dawley rats were employed in the study to reveal the role of HO-1 in various forms of kidney disease. This rat model showed characteristic renal and extrarenal phenotypes. These rats showed hemolytic anemia with morphological abnormality of the circulating erythrocytes, such as poikilocytosis and the presence of target cells and acanthocytes. Splenomegaly, leukocytosis and impaired growth were also observed. Most rats died early, by 6 months old. Whether the observed splenomegaly was transient or would result in progressive atrophy (as shown in the mouse model of Kovtunovych et al. [29]) was not clear from their data, because the study did not extend beyond 24 weeks old. Renal dysfunction with structural abnormality and proteinuria was another characteristic finding. Glomerular damage was characterized by increased mesangial matrix and lesions of focal and segmental sclerosis. Electron microscopy of the glomeruli revealed edematous podocytes with scant areas of foot process effacement. In contrast to human cases or mice models, iron deposition was not increased within the tubular epithelium or hepatic parenchyma of the rat model. In contrast to the progressive iron deposition in macrophages within the splenic red pulp, little iron deposition was observed even in older HO-1-deficient rats.

Table 3 summarizes the characteristics of HO-1-targeted mice in comparison with 8 human cases of HO-1 deficiency. In addition to intense inflammation and hemolytic anemia, abnormal iron deposition within renal tubular cells, and hepatic parenchymal cells, elevated levels of ferritin were also commonly observed in both mouse models and human cases. Although the spleen was detectable or enlarged in two of the human cases and in mouse models, asplenia or abnormal splenic pathology may also be a hallmark of HO-1 deficiency.

Based on reports describing human cases of HO-1 deficiency and animal models of HO-1 gene knockout, the spleen seems to show two potential fates in the absence of functional HO-1 protein. One extreme is the loss of splenic tissue early in utero. In these cases, the spleen is undetectable at birth, as seen in the first 6 cases of human HO-1 deficiency. The other extreme is progressive splenomegaly after birth and later atrophy of the spleen secondary to fibrosis of the red pulp (Figure 4).

## 4. HO-1 Deficiency and Oxidative Stress

Pathological examination of the first case of HO-1 deficiency revealed characteristic tissue injury. Of particular interest was the fact that cell injury was not observed ubiquitously, but instead confined to selected organs or specific cell types. Among these, the kidneys, liver, circulating monocytes, and vascular endothelial cells are the tissues showing apparent injury and dysfunction [31]. In the first human case, mild mesangial proliferation and thickening of the capillary loop were observed within the glomeruli. Electron microscopy revealed marked swelling and detachment of endothelial cells throughout the glomerular capillary. In addition to the glomerular damage, tubulointerstitial injury with tubular atrophy was significant. The liver was massively enlarged and significant amyloid accumulation was evident, resulting in marked atrophy of hepatocytes. Scattered foci of iron deposits were observed in both the kidneys and liver. The cytoplasm of circulating monocytes was vacuolated and monocyte surface antigens were significantly different from normal profiles.

This selective organ damage in HO-1 deficiency seems to have several causes. First, these susceptible cells are constantly exposed to oxidative stress, for both anatomical and functional reasons. Vascular endothelial cells are the recipients of shear stress and are exposed to multiple oxidative stressors, including hemolysis, pH changes, and hypoxemia. In the HO-1 deficiency patient, secondary accumulation of heme proteins, cholesterol and fragmented erythrocytes further aggravates these oxidative stresses. Renal tubular cells are constantly exposed to hematuria, proteinuria and various other substances excreted in urine. Resident macrophages and circulating monocytes are frequently activated to perform scavenger functions, and also function as central players in both innate and acquired immunity.

Second, these cells may be particularly sensitive to oxidative injury, but at the same time, may also serve as a high-quality sensor of oxidative stress for the organs. Oxidative insults to these susceptible cells rapidly induce release of anti-oxidative molecules, saving the organs from critical damage.

Last, these cells, because they have to serve as oxidative stress sensors, have some capacity for neutralization to combat these insults. HO-1 may play roles as a central molecule in all these mechanisms.

Another distinct feature of the first case of human HO-1 deficiency was the defective endothelial function, as represented by extremely abnormal parameters for coagulation and fibrinolysis. Compared to other hematological illnesses associated with disseminated intravascular coagulation, the patient exhibited extraordinarily elevated values for thrombin-antithrombin complex, fibrinogen degradation products, and palsmin-α_2_ plasmin inhibitor complex. Paradoxically, platelet numbers were constantly elevated. The data indicated that HO-1 or HO-1 products, such as CO, may be associated with regulation of the coagulation/fibrinolytic system. We recently demonstrated in in vitro cultures that a CO-releasing molecule, tricarbonyldichlororuthenium (II) dimer suppressed TNF-α-induced up-regulation of tissue factor and plasminogen activator inhibitor type 1 by human umbilical vein endothelial cells. The CO-releasing molecule also suppressed activation of mitogen-activated protein kinase and nuclear factor-kappa B signaling pathways by TNF-α. LPS-induced TNF-α production by circulating mononuclear cells was also significantly inhibited by the CO-releasing molecules [32]. These results may well explain the characteristic findings seen in HO-1 deficiency patients. At the same time, these findings support the view that CO-releasing molecules may provide a novel anti-coagulative and anti-inflammatory therapy [33,34].

A summary of macrophage activation and endothelial cell dysfunction is shown in Figure 5. The lack of HO-1 resulted in unregulated activation of macrophages and subsequent release of excessive inflammatory cytokines. At the same time, HO-1 deficiency resulted in overproduction of tissue factor by endothelial cells, leading to abnormal activation of the coagulation/fibrinolysis system. Figure 5 illustrates the role of HO-1 as an inhibitor of cytokine overproduction and endothelial cell dysfunction associated with catastrophic tissue injury seen in systemic inflammatory response syndrome.

## 5. Novel Anti-Inflammatory Therapies

Monocytes/macrophages comprise at least two functionally distinct subsets, M1 and M2 [35,36,37]. The different subsets of monocyte/macrophage lineage differentiate in response to environmental stimuli. M1 macrophages are the “classical” macrophages, representing the pro-inflammatory subset, whereas M2 macrophages are “alternatively” activated macrophages that resolve inflammatory responses, perform scavenger functions and promote tissue remodeling and repair. Interferon-γ is the key cytokine driving the M1 pathway, whereas IL-4, IL-10, and steroids promote monocyte differentiation into the M2 lineage [38,39].

We have previously reported that circulating monocytes produce significant levels of HO-1 during Kawasaki disease and infectious diseases, suggesting a certain anti-inflammatory role during inflammatory illnesses [40]. Furthermore, we investigated the profiles of cytokine mRNA expression in two subsets of circulating monocytes [41]. In that study, freshly isolated CD16^high^ CCR2^negative^ monocytes expressed significant levels of HO-1 mRNA in vivo. They produced little IL-10 on stimulation with LPS. In contrast, a major subset of CD16^low^ CCR2^positive^ monocytes did not express HO-1 mRNA in vivo, but responded significantly to LPS and produced IL-10. The fractions of CD16^high^ CCR2^negative^ monocytes increased during various acute inflammatory diseases, such as Kawasaki disease and influenza virus infection, suggesting the anti-inflammatory roles played by monocytes through HO-1 production.

In macrophages and dendritic cells, CO reduces proinflammatory and increases anti-inflammatory cytokine secretion in response to LPS [42,43]. HO-1-mediated anti-inflammatory effects may therefore be closely linked to anti-inflammatory mechanisms, such as the suppression of immune and inflammatory responses in macrophages via diminished antigen-presenting capacity and cytokine synthesis [44,45]. This is consistent with our finding in HO-1 deficiency patients, in whom the lack of HO-1 resulted in a marked rise in circulating heme and subsequent oxidative vascular and tissue injuries, anemia, and chronic inflammation.

HO-1 is induced by CD163-mediated uptake of the Hb-Hp complex [46,47,48]. Schaer et al. reported that macrophages express upregulated levels of CD163 in sepsis-induced hemophagocytic syndrome [49]. These macrophages expressed significant levels of HO-1, suggesting their role as a negative regulator of inflammation. We demonstrated that serum HO-1 concentrations correlate closely with serum sCD163 concentrations, and these are extremely high in the macrophage activation syndrome associated with sJIA [50]. These findings indicate that serum HO-1 might be derived from CD163^+^ alternatively activated macrophages. In particular, hemophagocytic macrophages may represent a major source of HO-1 in sJIA. Increased levels of serum HO-1, as well as CD163, suggest that alternative activation of macrophages is switched on in sJIA.

For many years, various anti-inflammatory agents, including steroids, non-steroidal anti-inflammatory drugs (NSAIDs), and immunosuppressants have been the main agents used to control excessive inflammation and tissue/organ injury. These agents are effective in some cases, but conventional anti-inflammatory/immunosuppressive therapies resulted in therapeutic failure in HO-1 deficiency cases for two reasons. One important and critical reason is that inflammation and immune reactions are, despite causing extensive tissue injury, organ damage, and discomfort to the patient, the critical components of host responses to pathogens and insults. Non-judicious use of conventional drugs may result in the host failing to expel pathogens and instead succumbing to overwhelming infection or uncontrollable cancer growth. Second, when the tissues and organs are already significantly damaged after the pathogen insult and subsequent host immune responses, addition of anti-inflammatory or immunosuppressive agents may further aggravate the host status. Understanding the built-in mechanism protecting cells and organs from oxidative injury, I would like to propose the third alternative of anti-inflammatory therapy to promote the anti-oxidative functions of each cell.

The prime target of such therapy is HO-1. Pharmacological induction of cellular HO-1 production, use of CO-releasing molecules, or enhancement of Hb-Hp receptor expression by steroid are examples of these approaches. As Naito et al. recently proposed, several pharmacological agents can drive a phenotypic shift to M2 macrophages and enhanced HO-1 production in vivo [51]. In a recent review on haptoglobin, di Masi et al. stress the anti-inflammatory role of haptoglobin in various clinical settings. Hp not only regulates Hb clearance from circulation, but at the same time modulates the immune status of the host [52]. Considering the role of haptoglobin as the modulator of inflammation, it becomes a potential target of anti-inflammatory therapy.

The disease caused by SARS-CoV-2 is becoming a common burden worldwide as the numbers of the patients increase progressively. Although the most patients recover without significant complication, small, but significant proportion of the patients with COVID-19 experience a broad array of organ damage. Accumulating evidence suggest that the major pathology of COVID-19 is characterized not only by acute respiratory distress syndrome, but at the same time excessive systemic inflammatory reaction and coagulopathy [53]. Considering these unique pathology of COVID-19, pharmacological induction of HO-1 in vivo may certainly serve as a potential therapeutic target to overcome the severe complication of COVID-19 [54,55,56].

## 6. Conclusions

Collective data obtained from 9 human cases of HO-1 deficiency and HO-1-knockout animal models support the view obtained from in vitro experiments. HO-1 plays cardinal roles in abrogating oxidative stress and inflammation, thus ameliorating cell and tissue injuries. Pharmacological manipulation of HO-1 production may provide an effective intervention to regulate inflammation and protect organs without introducing immunosuppression or cytocidal agents.

Furthermore, recognition of denominators for HO-1 deficiency will help in identifying this extremely rare, underdiagnosed disease at an early stage. Avoidance of exogenous stress along with appropriate treatment may prevent early death in these patients.

## Figures and Tables

**Figure 1 ijms-22-01514-f001:**
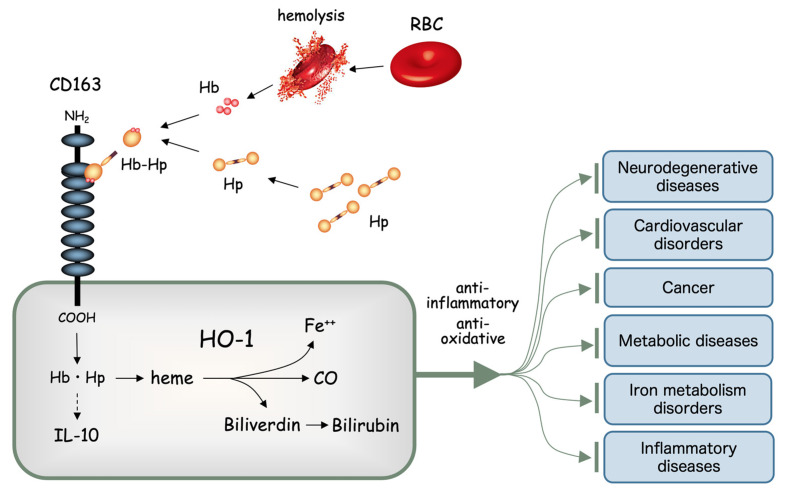
Heme oxygenase (HO)-1 and the heme degradation pathway. Free hemoglobin (Hb) quickly binds with haptoglobin (Hp) to form a molecular complex (Hb-Hp). Hb-Hp is rapidly incorporated through the scavenger receptor, now known as CD163, expressed on the cell surface of tissue macrophages such as hepatic Kupffer cells. Heme derived from Hb is degraded by HO-1 into free iron (Fe^++^), carbon monoxide (CO) and biliverdin. Biliverdin is further degraded into bilirubin by biliverdin reductase. All these heme degradation products are known to exert potent anti-inflammatory and anti-oxidative stress functions, which in turn play significant roles in preventing various chronic diseases.

**Figure 2 ijms-22-01514-f002:**
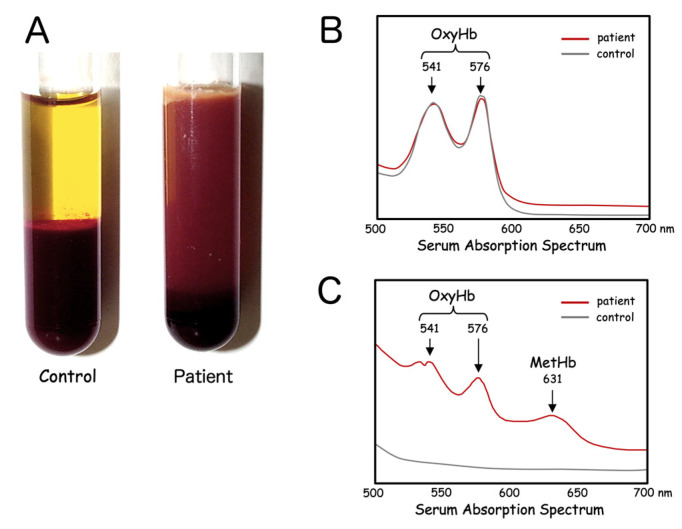
Serum characteristics of HO-1 deficiency. Patient serum separated immediately after blood drawing is characterized by a turbid, dark-brown color (**A**). Fresh hemolysate of the patient’s erythrocytes shows only the peaks of oxyhemoglobin (OxyHb) (**B**). Absorption spectrum of the patient’s serum reveals inclusion of large amounts of both OxyHb and methemoglobin (MetHb) (**C**).

**Figure 3 ijms-22-01514-f003:**
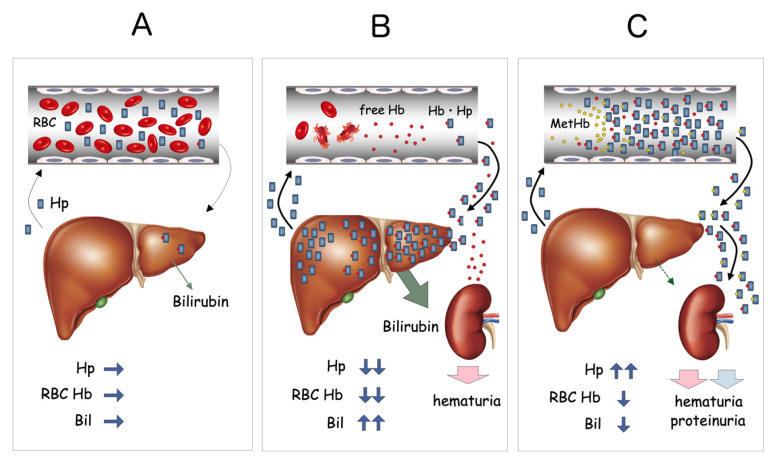
Proposed mechanism of MetHb accumulation in blood. Under normal conditions, small amounts of Hb derived from hemolysis are rapidly bound by circulating Hp, forming Hb-Hp complexes, which are rapidly taken up by the liver through CD163 (**A**). When massive hemolysis occurs, as in the case of hemolytic anemia, large amounts of free Hb are bound by serum Hp and transferred to the liver (**B**). This results in a significant reduction of serum Hp and increased indirect bilirubin (Bil) derived from the heme degradation pathway. In HO-1-deficient patients, both OxyHb and MetHb bound to Hp accumulate within the serum. Despite the apparent intravascular hemolysis, total Hp content is significantly increased, serum bilirubin level remains low, and overflow of Hb-Hp complex into the urine is seen (**C**).

**Figure 4 ijms-22-01514-f004:**
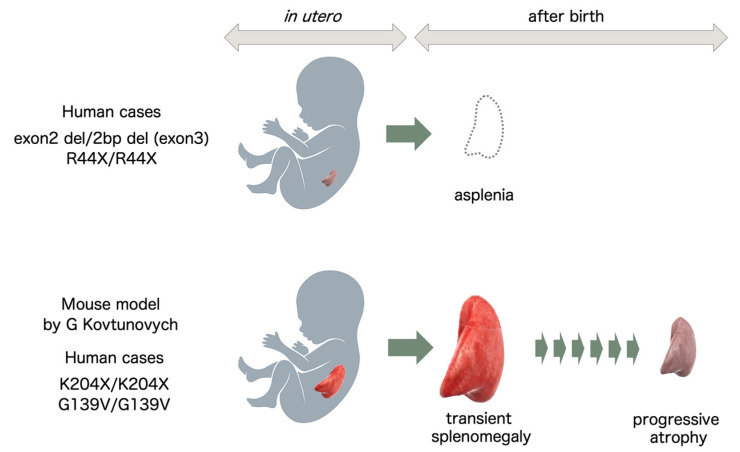
HO-1 deficiency and splenic development. In human cases of HO-1 deficiency with mutations found in Japan and India, the patients were found to be asplenic, presumably reflecting intrauterine defects in splenic development. In mouse models of HO-1 deficiency and human cases with mutations found in Turkey and Iran, the spleens showed transient splenomegaly followed by progressive atrophy after birth.

**Figure 5 ijms-22-01514-f005:**
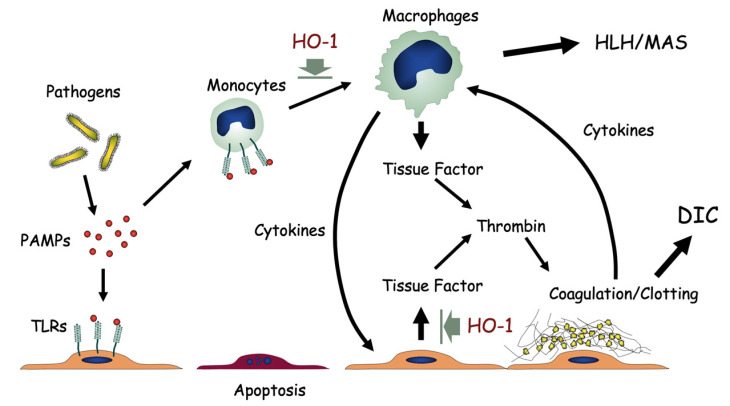
HO-1 deficiency and oxidative stress. Pathogen-associated molecular patterns (PAMPs) bind toll-like receptors (TLRs) on vascular endothelial cells or monocytes. Lack of HO-1 results in unregulated activation of macrophages with excess release of inflammatory cytokines. Hemophagocytic lymphohistiocytosis (HLH) or macrophage activation syndrome (MAS) may result. At the same time, HO-1 deficiency results in overproduction of tissue factors by endothelial cells, leading to abnormal activation of the coagulation/fibrinolysis system, and disseminated intravascular coagulation (DIC). Prolonged and sustained activation of monocytes, platelets and endothelial cells leads to exhaustion and dysfunction of these cells.

**Table 1 ijms-22-01514-t001:** Clinical profiles of human HO-1 deficiency with null mutations of *HMOX1.*

Findings	Case 1	Case 2	Case 3	Case 4	Case 5	Case 6	Case 7	Case 8	Case 9
Sex	male	female	male	male	female	male	female	male	male
Age of onset	2 yr	15 yr	6 m	10 yr	7 yr	13 m	17 m	3 m	4 yr
Age at diagnosis	5 yr	16 yr	2 yr	10 yr	9 yr	20 m	3 yr	20 m	10 yr
Fever	+	+	+	+	+	+	+	+	+
Arthralgia	+	-	-	-	-	-	-	-	+
Hemolytic anemia	+++	++	++	++	++	++	++	++	++
Jaundice	-	-	-	-	-	-	-	-	-
Hematuria/ proteinuria	+/+	+/+	+/+	+/+	+/+	+/+	?	?	+/+
Asplenia	+	+	+	+	+	+	-	splenomegaly	hyposplenia
Prominent forehead	+	+	+	+	+	+	?	?	+
Hypertension	+	+	+	+	+	-	-	-	-
Cerebral bleeding	+	+	+	-	-	-	-	-	-
Growth delay	+	-	+	+	+	+	+	?	+

+: present, -: absent, +++: strongly positive, ++: moderately positive, ?: unknown or not described.

**Table 2 ijms-22-01514-t002:** Laboratory data of human HO-1 deficiency with null mutations of *HMOX1.*

Findings	Case 1	Case 2	Case 3	Case 4	Case 5	Case 6	Case 7	Case 8	Case 9
Mutations of *HMOX1*	exon 2 del 2-bp del (exon 3)	R44X (homo)	R44X (homo)	R44X (homo)	R44X (homo)	R44X (homo)	K204X (homo)	G139V (homo)	c.636+2T>A p.L89Sfs24
Mother Father	exon 2 del exon 3; 2-bp del	not done not done	not done not done	not done not done	R44X/W * R44X/W	R44X/W R44X/W	K204X/W K204X/W	G139V/W G139V/W	c.636+2T>A/W p.L89Sfs24/W
CRP (mg/dL)	6.7	30.8	5.3	normal	24.0	4.8	12.4	24.0	not shown
FDP-DD (mg/mL)	186.1	>8	not done	normal	not done	not done	not done	not done	>20
WBC (×10^3^/mL)	51.6	18.5	38.0	39.6	not shown	43.2	33.0	19.9	53.8
Plt (×10^4^/mL)	226	137	109	117	100	123	100	47.8	91.4
Ferritin (ng/mL)	780	4912	15,530	15,358	2500	>2000	27,425	4855	1980
LDH (IU/L)	17,470	9462	12,858	16,000	4000	21,400	15,350	15,713	19,706
AST/ ALT (IU/L)	448/ 74	982/ 149	1080/ 283	689/ 68	300/ 80	652/ 133	580/ 813	not shown	301/ 74
Bilirubin (mg/dL)	0.2	0.64	1.2	0.3	0.4	0.02	0.8	0.2–1.6	0.2

* W; wild type.

**Table 3 ijms-22-01514-t003:** Mouse models and human cases of HO-1 deficiency.

Findings	Mouse Models		Human Cases				
	Poss et al. [25,26]	Kovtunovych et al. [27]	Japanese Case	Indian Cases	Iranian Case	Turkish Case	USA Case
Gene mutations	knockout	knockout	exon 2 del 2-bp del (exon 3)	R44X (homo)	K204X (homo)	G139V (homo)	c.636 + 2T>A p.L89Sfs24
Type of mutations	large deletion	large deletion	nonsense mutation	nonsense mutation	nonsense mutation	missense mutation	nonsense mutation
Anemia	+	+	+	+	+	+	+
RBC anomaly	?	+	+	+	?	+	+
Spleen	splenomegaly	splenomegaly → atrophy	asplenia	asplenia	normal size	splenomegaly	hyposplenia
Inflammation	++	++	++	++	++	++	++
Iron deposit	++	++	++	?	++	?	++
Low bilirubin	?	?	yes	yes	yes	yes	yes
Ferritin	high	high	high	high	high	high	high
LDH	?	?	high	high	high	high	high
Platelet	?	increased	increased	increased	increased	normal	increased

+: present, -: absent, ++: moderately positive, ?: unknown or not described.

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
