# Peer review of "Heme Oxygenase-1 Deficiency and Oxidative Stress: A Review of 9 Independent Human Cases and Animal Models"

_ijms, 2021, doi:10.3390/ijms22041514_

Round 1
Reviewer 1 Report
The paper by Yachie is fairly and certainly deserves publication, even though I'm wondering whether, due to its strong clinical character, International Journal of Molecular Sciences is the most appropriate journal for this. Probably, International Journal of Clinical Medicine would be more appropriate.
Furthermore, I found that the literature references are a little too old. There are many interesting papers and reviews in the last two years, which are not quoted and they should. Just few examples are:
Chau et al. (2020) Pediatr. Rheumatol. Vol. 18, 80, which is dealing with the connection between heme oxygenase and interstitial pneumonia
di Masi et al. (2020) Mol. Asp. Med. Vol. 73, 100851, which is a very comprehensive review of the role of haptoglobin from many viewpoint
Hooper (2020) in "Cell Stress and Chaperones", which is connecting the role of heme oxygenase and COVID-19.
In this respect, author should add a chapter commenting on the evidences of the role of heme oxygenase in the very recent COVID-19 pandemia. I think that this point must be dealt with in this manuscript
Author Response
Dear Sir
I appreciate your helpful comments and suggestions.
In particular, I truly thank you for mentioning the 9th human case of HO-1 deficiency, which I missed to pick up.
I understand that my manuscript is largely clinical and may not be suitable for this journal. However, the clinical description of the patients and animal models will certainly contribute to the understanding of the molecular role of heme oxygenate and related molecules, I believe.
I intentionally avoided to mention COVID-19 initially, because that was beyond the perspective of the current manuscript. However, I added some phrased in the last part of the manuscript, referring the related papers.
In the revised manuscript,
1) I added the 9th case of human HO-1 deficiency with the reference.
2) I revised tables and the main text accordingly.
3) I rewrote the final paragraph, according to the reviewer's comment.
Please review the manuscript again and consider this for the publication in this journal.
With best regards,
Akihiro Yachie

Reviewer 2 Report
Heme oxygenase-1 deficiency and oxidative stress: a review of 8 independent human cases and animal models
In general, the manuscript is well organized and written and provides a valuable summary of human clinical cases with HO-1 deficiency.
Since the author is the first one to identify genetic deficiency of HO-1 in human case and experienced a series of investigations of the clinical characteristics, the author may provide more results or suggestions with some feasible therapy strategies with the deficiency.
Minor concerns
- Figure 1, some parts of the bottom are cut off.
- Line 89-90, “the spleen was not detected”, the expression is not clear, does it mean “ the enlargement was not detected in the spleen?”
- Line 144, Further studies…
-
Author Response
Dear Sir
I appreciate your kind review of the manuscript.
Followings are my response to the comments.
1) Figure 1; The lower part of the Figure 1 was not seen initially, but it should be seen now after the editorial correction.
2) About Spleen; I used the word "absent" to make it clear.
3) Further studies; I could not read the following words. However, I deleted the last sentences of paragraph 5. from the text.
4) According to the reviewer 1's comments, I added Case 9 to the lists of human HO-1 deficiency and changed the test accordingly. Furthermore, I added phrased to mention the potential role of HO-1 in the treatment of COVID-19.
Please review the revised manuscript and consider this for the publication in the journal.
With best regards,
Akihiro Yachie

Reviewer 3 Report
This review clearly describes the clinical features of HO-1 deficient patients. The thinking process when Dr. Yachie first reports a HO-1 deficient patient is also helpful to clinicians who may be treating other unknown diseases. In recent years, the relationship between HO-1 and pathological conditions has been reported in various diseases such as infectious diseases, arteriosclerosis, and systemic lupus erythematosus. It is expected that HO-1 will be clinically applied as a new therapeutic target, and I think this review article promotes the understanding of researchers and clinicians in this field.
Author Response
Dear Sir
I appreciate your kind review of the manuscript.
According to the reviewer 1's comments, I added Case 9 to the lists of human HO-1 deficiency and changed the test accordingly. Furthermore, I added phrased to mention the potential role of HO-1 in the treatment of COVID-19.
Please review the revised manuscript and consider this for the publication in the journal.
With best regards,
Akihiro Yachie

Round 2
Reviewer 1 Report
I consider the present revised version acceptable for publication